# The clinical effectiveness of REGEN-COV in SARS-CoV-2 infection with Omicron versus Delta variants

Hayley B. Gershengorn[1,2]*, Samira Patel[3], Tanira Ferreira[1], Sankalp Das[3], Dipen J. Parekh[4], Bhavarth Shukla[5]

1 Division of Pulmonary, Critical Care, and Sleep Medicine, Department of Medicine, University of Miami Miller School of Medicine, Miami, FL, United States of America, 2 Division of Critical Care Medicine, Albert Einstein College of Medicine, Bronx, NY, United States of America, 3 Care Transformation, University of Miami Hospital and Clinics, Miami, FL, United States of America, 4 Department of Urology, University of Miami Miller School of Medicine, Miami, FL, United States of America, 5 Division of Infectious Diseases, Department of Medicine, University of Miami Miller School of Medicine, Miami, FL, United States of America

* hbg20@med.miami.edu

**Data Availability Statement:** Data cannot be shared publicly because of HIPAA. Data are available from the University of Miami Institutional Data Access / Ethics Committee (contact via

## Abstract

### Background

In vitro studies suggesting that REGEN-COV (casirivimab plus imdevimab monoclonal antibodies) had poor efficacy against Omicron-variant SARS-CoV-2 infection led to amendment of REGEN-COV's Emergency Use Authorization to recommend use only in regions without high Omicron prevalence. REGEN-COV's relative clinical effectiveness for Omicron is unknown.

### Methods and findings

We conducted a retrospective cohort study of non-hospitalized adults who tested positive for SARS-CoV-2 by polymerase chain reaction at the University of Miami Health System from July 19–November 21, 2021 (Delta period) and December 6, 2021–January 7, 2022 (Omicron period). Subjects were stratified be REGEN-COV receipt within 72h of test positivity and by time period of infection. We constructed multivariable logistic regression models to assess the differential association of REGEN-COV receipt with hospitalization within 30 days (primary outcome) and ED presentation; all models included three exposure terms (REGEN-COV receipt, Omicron vs Delta period, interaction of REGEN-COV with time period) and potential confounders (vaccination status, vaccine boosting, cancer diagnosis). Our cohort consisted of 2,083 adults in the Delta period (213 [10.2%] received REGEN-COV) and 4,201 in the Omicron period (156 [3.7%] received REGEN-COV). Hospitalization was less common during the Omicron period than during Delta (0.9% vs 1.7%, p = 0.78) and more common for patients receiving REGEN-COV than not (5.7% vs 0.9%, p<0.001). After adjustment, we found no differential association of REGEN-COV use during Omicron vs Delta with hospitalization within 30d (adjusted odds ratio [95% confidence interval] for the interaction term: 2.31 [0.76–6.92], p = 0.13). Similarly, we found no differential association for hospitalization within 15d (2.45 [0.63–9.59], p = 0.20) or emergency department presentation within 30d (1.43 [0.57–3.51], p = 0.40) or within 15d (1.79 [0.65–4.82], p = 0.30).

hsro@miami.edu) for researchers who meet the criteria for access to confidential data.

**Funding:** All authors (HBG, SP, TF, SD, DJP, BS) receive support from the University of Miami Hospital and Clinics Data Analytics Research Team (UHealth-DART) of which all are members. The funder had no role in study design, data collection and analysis, decision to publish, or preparation of the manuscript.

**Competing interests:** HBG was a member of a scientific advisory board for COVID therapeutics for Gilead Sciences, Inc. The other authors have declared that no competing interests exist. This does not alter our adherence to PLOS ONE policies on sharing data and materials.

## Conclusions

Within the limitations of this study's power to detect a difference, we identified no differential effectiveness of REGEN-COV in the context of Omicron vs Delta SARS-CoV-2 infection.

## Introduction

REGEN-COV, the cocktail of monoclonal antibodies casirivimab and imdevimab, was given Emergency Use Authorization (EUA) for the treatment of patients with mild-moderate COVID-19 at high risk of progression to severe disease by the U.S. Food and Drug Administration (FDA) on November 21, 2020 [1]. A phase 3 study enrolling patients through February, 2021 demonstrated significant reductions in COVID-19-related hospitalization or all-cause mortality by day 29 in this population [2]. By the winter of 2021/2022, however, as the Omicron variant began to predominate in the U.S., in vitro studies suggested that REGEN-COV would have less efficacy in this context [3–7]. On January 24, 2022, the FDA amended the REGEN-COV EUA "to exclude geographic regions where. . . infection or exposure is likely due to a variant that is non-susceptible to REGEN-COV" [1].

In South Florida, new cases of COVID-19 were very low at the time Omicron arrived [8], allowing for clear demarcations of a Delta (summer, 2021) and a separate Omicron (winter, 2021–22) wave. In the University of Miami Hospital Health System (UHealth), Delta accounted for >90% of samples tested by quantitative polymerase chain reaction (PCR) from mid-July through the third week of November, while Omicron represented >90% by the second week of December. In this context, we sought to examine the clinical effectiveness of REGEN-COV administration for outpatients presumed to have Delta versus Omicron infection. We hypothesized that REGEN-COV would be less effective against Omicron SARS-CoV-2.

## Patients and methods

We conducted a retrospective cohort study of non-hospitalized adults who tested positive for SARS-CoV-2 by PCR at UHealth from July 19 –November 21, 2021 (Delta period) and December 6, 2021 –January 7, 2022 (Omicron period). We excluded subjects tested after January 7, 2022 as this was the last date REGEN-COV was prescribed for any patient at UHealth (suggesting non-availability thereafter). Subjects were also excluded if they were <18 years-old, had a positive SARS-CoV-2 test in our system within 90d prior to index testing, or presented to the emergency department (ED) or were hospitalized within 72h following testing (to mitigate against immortal time bias).

Subjects were stratified into those who received REGEN-COV (casirivimab 600mg / imdevimab 600mg intravenously once) within 72h of testing positive and those who did not; [2] we initially planned to evaluate 7d from testing as a surrogate for time since symptom onset (as was used in the trial), but all but 3 patients who met our primary outcome (see below) received REGEN-COV within 72h (see below). Subjects who received REGEN-COV after 72h were considered to have not received it. Notably, there was no access to Sotrovimab at UHealth during this period; similarly, Remdesivir was not available for administration to outpatients and neither nirmatrelvir/ritonavir nor molunipiravir were prescribed for any cohort patients. Our primary outcome was hospitalization at UHealth within 30d of positive testing. We also evaluated secondary outcomes of hospitalization with 15d and ED presentation at UHealth within 30d and 15d.

## Statistical analysis

We described our cohort stratified by time period and REGEN-COV receipt using standard summary statistics. Comparisons between groups were made using Chi-square, Fisher's exact, and Wilcoxon rank-sum testing as appropriate.

We then conducted multivariable logistic regression models to assess the differential impact of REGEN-COV on our outcomes, whether subjects tested positive in the Delta or Omicron periods. These models included three exposure terms—time period (Delta vs Omicron), REGEN-COV receipt, and their interaction; they also included vaccine receipt (not fully vaccinated, fully vaccinated <6 months prior to testing, full vaccinated ≥6 months prior to testing), vaccine booster receipt, and cancer history (based on International Classification of Diseases, 10[th] Revision [ICD-10] coding in UHealth system for leukemia, lymphoma, or solid tumor as defined for the Charlson comorbidity index [9]). In these models, a p-value <0.05 for the interaction term indicates a differential association of REGEN-COV with the outcome in the Omicron (vs the Delta) period.

We conducted several post hoc sensitivity analyses to address issues of potential residual confounding and misclassification bias. To better address confounding by illness severity, we first included an additional covariable for whether each patient had any ICD-10 codes in our system (suggesting knowledge of their medical history). Second, we performed 1:4 propensity score matching of patients who received REGEN-COV and those who did not in, separately, the Delta and Omicron periods. Finally, we created an overfit model adding covariables for demographics and more available medical history for the full cohort and then limited to those patients in whom we had at least one ICD-10 code (again, indicating knowledge of their history). To address concerns about exposure and outcomes misclassification, we limited our cohort to patients who had either a hospitalization or an ED visit to UHealth within the prior year. This analysis sought to focus on patients most likely to receive all their care at UHealth and who would, therefore, but more likely to get REGEN-COV and be hospitalized in our system, if at all.

All analyses were performed using R (R Foundation for Statistical Computing, Vienna, Austria) and Excel (Microsoft Office 365, Redmond, WA). This project was approved by the University of Miami Institutional Review Board (#20200739) and a waiver of informed consent was granted.

## Results

Our primary cohort consisted of 2,083 adults in the Delta period (213 [10.2%] received REGEN-COV) and 4,201 in the Omicron period (156 [3.7%] received REGEN-COV; Table 1). Patients receiving REGEN-COV were older (median [interquartile range, IQR]: 59 [46–70] vs 36 [24–51], p<0.001), more often insured by Medicare (14% vs 3.5%, p<0.001), and more commonly with chronic conditions (67% vs 29%, p<0.001).

### Primary outcome: Hospitalization within 30 Days

Hospitalization within 30d was more common for patients receiving REGEN-COV (5.7% vs 0.9%, p<0.001). Hospitalization was less common during the Omicron time period than during Delta (0.9% vs 1.7%, p = 0.78; Table 2).

After adjusting for vaccination, vaccine boosting, and cancer status, we found no differential association of REGEN-COV with hospitalization at 30d during Omicron vs Delta (adjusted odds ratio [aOR]: 2.31 [95% confidence interval, CI]: 0.76–6.92, p = 0.13; Table 3). However, there was a non-statistically significant increased odds of hospitalization with REGEN-COV receipt (1.95 [0.86–4.18], p = 0.094) and a similarly non-statistically significantly decreased odds of hospitalization during Omicron (0.58 [0.32–1.05], p = 0.071). The marginal probability

**Table 1. Cohort characteristics by REGEN-COV receipt.**

| | Full Cohort[a] | No REGEN-COV[a] | REGEN-COV[a] | p-value |
|---|---|---|---|---|
| # of Patients, N (row %) | 6284 | 5915 (94.1%) | 369 (5.9%) | |
| Time Period | | | | <0.001 |
| Delta | 2,083 (33%) | 1,870 (32%) | 213 (58%) | |
| Omicron | 4,201 (67%) | 4,045 (68%) | 156 (42%) | |
| Demographics | | | | |
| Age, med(IQR) | 37 (24, 52) | 36 (24, 51) | 59 (46, 70) | <0.001 |
| Female | 3,717 (59%) | 3,519 (59%) | 198 (54%) | |
| Race/Ethnicity | | | | <0.001 |
| Non-Hispanic White | 1,248 (20%) | 1,176 (20%) | 72 (20%) | |
| Hispanic White | 2,277 (36%) | 2,096 (35%) | 181 (49%) | |
| Hispanic Black | 92 (1.5%) | 85 (1.4%) | 7 (1.9%) | |
| Non-Hispanic Black | 817 (13%) | 739 (12%) | 78 (21%) | |
| Other/Unknown | 1,850 (29%) | 1,819 (31%) | 31 (8.4%) | |
| Payor | | | | <0.001 |
| Commercial | 5,309 (84%) | 5,023 (85%) | 286 (78%) | |
| Medicaid | 34 (0.5%) | 30 (0.5%) | 4 (1.1%) | |
| Medicare | 257 (4.1%) | 207 (3.5%) | 50 (14%) | |
| Unknown/Other | 684 (11%) | 655 (11%) | 29 (7.9%) | |
| Vaccination Status | | | | |
| Vaccination Status | | | | <0.001 |
| Not Vaccinated | 2,479 (39%) | 2,297 (39%) | 182 (49%) | |
| Fully Vaccinated ≥ 6 Months | 1,850 (29%) | 1,779 (30%) | 71 (19%) | |
| Fully Vaccinated < 6 Months | 1,955 (31%) | 1,839 (31%) | 116 (31%) | |
| Boosted | 999 (16%) | 938 (16%) | 61 (17%) | 0.79 |
| Chronic Health Condition | | | | |
| History known | 3,799 (60%) | 3,496 (59%) | 303 (82%) | <0.001 |
| Cancer diagnosis | 350 (5.6%) | 268 (4.5%) | 82 (22%) | <0.001 |
| Any Elixhauser Comorbidity | 1,990 (32%) | 1,743 (29%) | 247 (67%) | <0.001 |
| # of Elixhauser Comorbidities, med(IQR) | 0 (0, 1) | 0 (0, 1) | 1 (0, 3) | <0.001 |
| Visited UHealth within prior 1 year | 1,460 (23%) | 1,140 (19%) | 320 (87%) | <0.001 |
| Outcomes | | | | |
| Hospitalized within 30d | 72 (1.1%) | 51 (0.9%) | 21 (5.7%) | <0.001 |
| Hospitalized within 15d | 42 (0.7%) | 28 (0.5%) | 14 (3.8%) | <0.001 |
| ED Presentation within 30d | 118 (1.9%) | 91 (1.5%) | 27 (7.3%) | <0.001 |
| ED Presentation within 15d | 80 (1.3%) | 56 (0.9%) | 24 (6.5%) | <0.001 |

d: days; ED: emergency department; IQR: interquartile range; med: median; UHealth: University of Miami Hospital and Clinics Health System

a percentages depict the fraction of each population (column percentages) unless otherwise indicated

of hospitalization remained higher among patients who received REGEN-COV in both time periods (5.2% in Delta vs 6.4% in Omicron) versus those who did not (1.3% in Delta vs 0.7% in Omicron; Fig 1). All sensitivity analyses produced similar results of no differential association of REGEN-COV with hospitalization at 30d during Omicron vs Delta (Fig 2, S1 Table).

## Secondary outcomes

Hospitalization within 15d was more common for patients receiving REGEN-COV (3.8% vs 0.5%, p<0.001) and less common during the Omicron vs Delta periods (0.5% vs 1.0%,

**Table 2. Outcomes by time period and REGEN-COV receipt.**

| | Delta Period | | | Omicron Period | | |
|---|---|---|---|---|---|---|
| | **All patients** | **No REGEN-COV** | **REGEN-COV** | **All patients** | **No REGEN-COV** | **REGEN-COV** |
| Hospitalization | | | | | | |
| within 30d | 35 (1.7%) | 24 (1.3%) | 11 (5.2%) | 37 (0.9%) | 27 (0.7%) | 10 (6.4%) |
| within 15d | 20 (1.0%) | 13 (0.7%) | 7 (3.3%) | 22 (0.5%) | 15 (0.4%) | 7 (4.5%) |
| ED Presentation | | | | | | |
| within 30d | 54 (2.6%) | 38 (2.0%) | 16 (7.5%) | 64 (1.5%) | 53 (1.3%) | 11 (7.1%) |
| within 15d | 40 (1.9%) | 26 (1.4%) | 14 (6.6%) | 40 (1.0%) | 30 (0.7%) | 10 (6.4%) |

d: days; ED: emergency department

p = 0.067). After multivariable adjustment, we found no differential association of REGEN-COV with hospitalization at 15d during Omicron vs Delta (aOR [95% CI]: 2.45 [0.63–9.59], p = 0.20).

ED presentation within 30d was more common for patients receiving REGEN-COV (7.3% vs 1.5%, p = <0.001) and less common during the Omicron vs Delta time (1.5% vs 2.6%, p = 0.005). After multivariable adjustment, there was no differential association of REGEN-COV with ED presentation within 30d during Omicron vs Delta (aOR [95% CI]: 1.43 [0.57–3.51], p = 0.40).

ED presentation within 15d was more common for patients receiving REGEN-COV (6.5% vs 0.9%, p = <0.001) and less common during the Omicron vs Delta time (1.0% vs 1.9%, p = 0.002). After multivariable adjustment, there was no differential association of REGEN-COV with ED presentation within 15d during Omicron vs Delta (aOR [95% CI]: 1.79 [0.65–4.82], p = 0.30).

## Discussion

We found no statistically significant differential association of REGEN-COV use within 72h of SARS-CoV-2 test positivity with the Delta vs Omicron variants and outcomes of either hospitalization or ED presentation within the subsequent month. However, across ten models, we consistently saw a nonsignificant increased odds of worse outcomes with REGEN-COV receipt in Omicron vs Delta and, in each, confidence intervals that were wide. These factors

**Table 3. Adjusted associations of time period, REGEN-COV, and the interaction with outcomes.**

| | Hospitalized 30d | | Hospitalized 15d | | ED Presentation 30d | | ED Presentation 15d | |
|---|---|---|---|---|---|---|---|---|
| **Model Variables** | **OR (95% CI)** | **p-value** | **OR (95% CI)** | **p-value** | **OR (95% CI)** | **p-value** | **OR (95% CI)** | **p-value** |
| Received REGEN-COV (vs not) | 1.95 (0.86, 4.18) | 0.09 | 2.47 (0.87, 6.47) | 0.08 | 3.75 (1.98, 6.82) | <0.001 | 4.72 (2.33, 9.20) | <0.001 |
| Omicron period (vs Delta) | 0.58 (0.32, 1.05) | 0.07 | 0.54 (0.24, 1.20) | 0.12 | 0.78 (0.50, 1.21) | 0.3 | 0.68 (0.39, 1.18) | 0.2 |
| Interaction (REGEN-COV*Omicron) | 2.31 (0.76, 6.92) | 0.13 | 2.45 (0.63, 9.59) | 0.2 | 1.43 (0.57, 3.51) | 0.4 | 1.79 (0.65, 4.82) | 0.3 |
| Vaccination Status | | | | | | | | |
| Not Vaccinated | reference | | reference | | reference | | reference | |
| Fully Vaccinated ≥ 6 Months | 0.68 (0.36, 1.24) | 0.2 | 0.88 (0.42, 1.82) | 0.7 | 0.46 (0.26, 0.76) | 0.004 | 0.35 (0.16, 0.69) | 0.004 |
| Fully Vaccinated < 6 Months | 0.44 (0.19, 0.93) | 0.045 | 0.19 (0.03, 0.65) | 0.025 | 0.55 (0.30, 0.94) | 0.036 | 0.63 (0.31, 1.17) | 0.2 |
| Boosted | 0.71 (0.24, 1.98) | 0.5 | 0.00 (0.00, 1.9E10) | >0.9 | 1.12 (0.55, 2.28) | 0.8 | 1.09 (0.47, 2.49) | 0.8 |
| Cancer diagnosis | 13.80 (8.22, 23.2) | <0.001 | 10.60 (5.28, 20.8) | <0.001 | 1.16 (0.57, 2.17) | 0.7 | 1.24 (0.55, 2.49) | 0.6 |

CI: confidence interval; d: days; OR: odds ratio

## A. Hospitalization within 30d

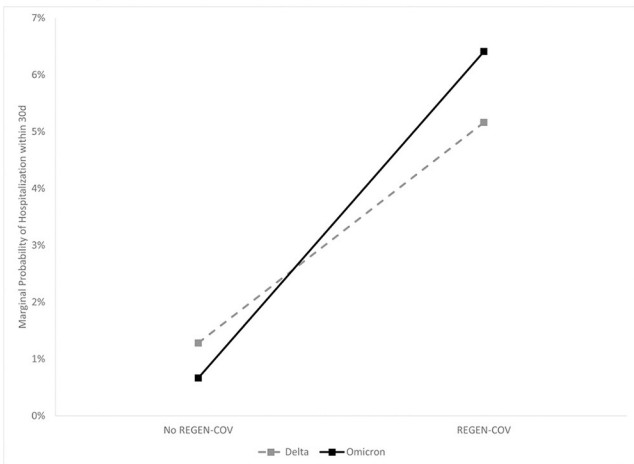

## B. Hospitalization within 15d

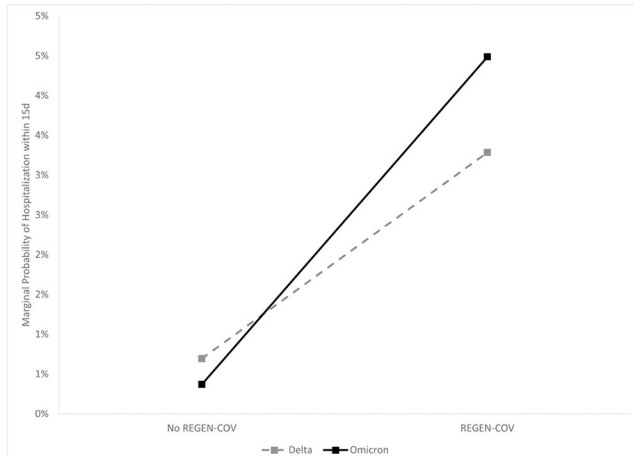

## C. ED Presentation within 30d

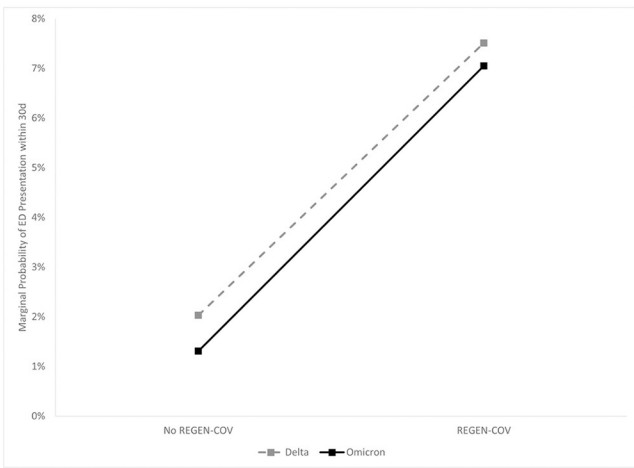

## D. ED Presentation within 15d

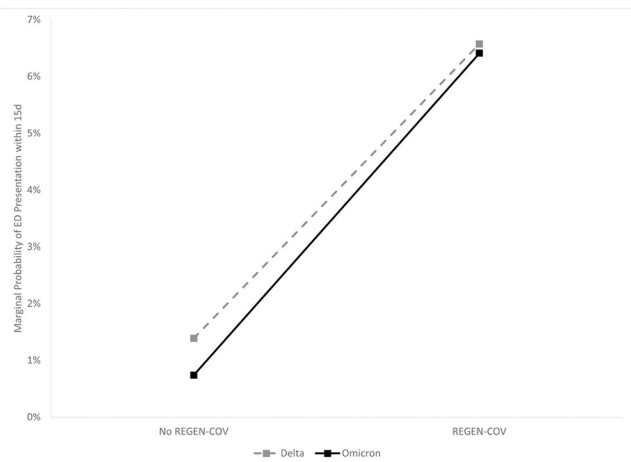

**Fig 1. Marginal probability of outcomes by time period and REGEN-COV receipt.** A. Hospitalization within 30d. B. Hospitalization within 15d. C. ED Presentation within 30d. D. ED Presentation within 15d. d: days; ED: emergency department.

together suggest our study may have been underpowered to detect a differential association that truly existed.

Our results have several potential explanations. First, there may truly be no relative ineffectiveness of REGEN-COV for infections with the Omicron as compared to the Delta variants. While in vitro studies suggest REGEN-COV should be less efficacious against the Omicron variant [3–7], a pre-clinical murine model demonstrated relatively preserved efficacy in vivo [10], consistent with our results. And, prior clinical evaluations assessing REGEN-COV's effectiveness across variants predated both Delta and Omicron emergence [11].

A second possible explanation for our findings of no differential association of REGEN-COV for Omicron vs Delta may be that our study was underpowered to detect differences that truly exist. This possibility is supported by the relatively wide confidence intervals surrounding our estimated odds ratios. As power for associative studies depends on the number of events, the fact that rates of severe outcomes overall (due to virulence differences [12], vaccination uptake, or something else) declined during the Omicron wave may partly explain our estimate imprecision. In this context, it is important to note that the point estimate in all models

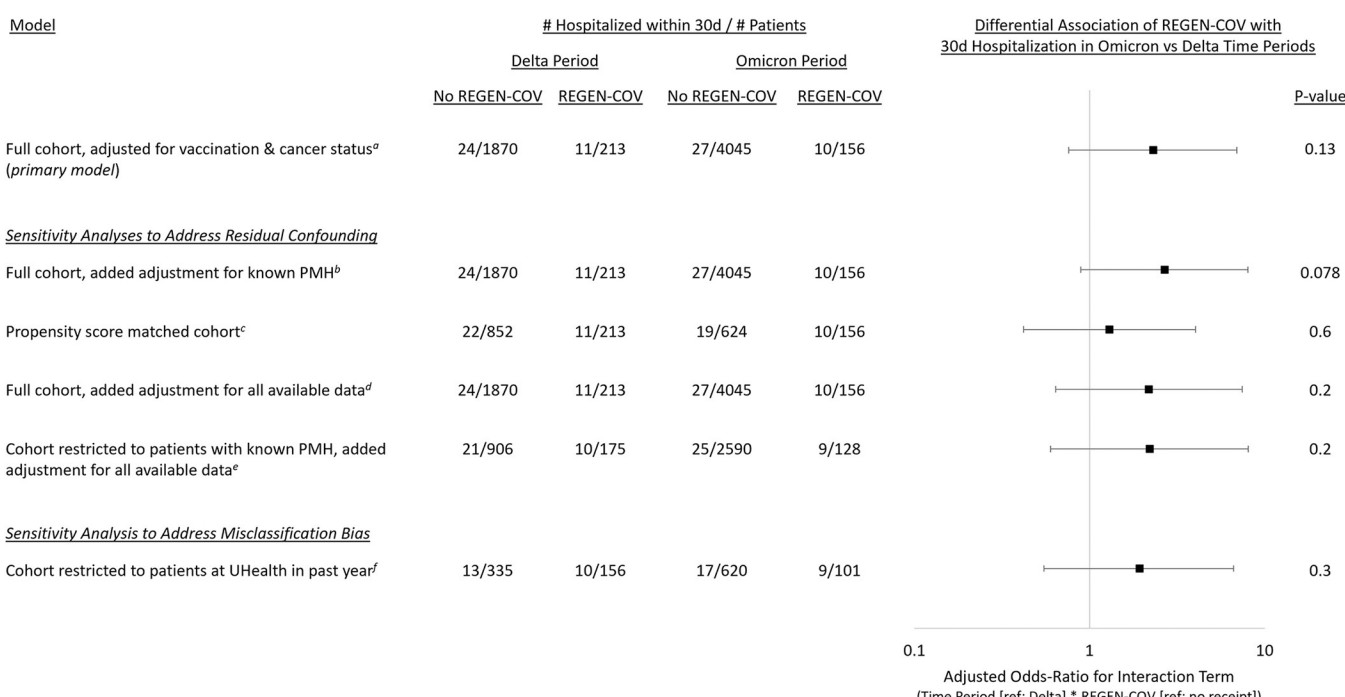

| Model | # Hospitalized within 30d / # Patients | | | | Differential Association of REGEN-COV with 30d Hospitalization in Omicron vs Delta Time Periods | |
| --- | --- | --- | --- | --- | --- | --- |
| | Delta Period | | Omicron Period | | | |
| | No REGEN-COV | REGEN-COV | No REGEN-COV | REGEN-COV | | P-value |
| Full cohort, adjusted for vaccination & cancer status[a] (*primary model*) | 24/1870 | 11/213 | 27/4045 | 10/156 | | 0.13 |
| *Sensitivity Analyses to Address Residual Confounding* | | | | | | |
| Full cohort, added adjustment for known PMH[b] | 24/1870 | 11/213 | 27/4045 | 10/156 | | 0.078 |
| Propensity score matched cohort[c] | 22/852 | 11/213 | 19/624 | 10/156 | | 0.6 |
| Full cohort, added adjustment for all available data[d] | 24/1870 | 11/213 | 27/4045 | 10/156 | | 0.2 |
| Cohort restricted to patients with known PMH, added adjustment for all available data[e] | 21/906 | 10/175 | 25/2590 | 9/128 | | 0.2 |
| *Sensitivity Analysis to Address Misclassification Bias* | | | | | | |
| Cohort restricted to patients at UHealth in past year[f] | 13/335 | 10/156 | 17/620 | 9/101 | | 0.3 |

0.1    1    10

Adjusted Odds-Ratio for Interaction Term
(Time Period [ref: Delta] * REGEN-COV [ref: no receipt])

**Fig 2. Differential association of REGEN-COV with hospitalization within 30 Days by time period.** d: days; PMD: primary medical doctor; PMH: past medical history; ref: reference. *a* This is the primary model including adjustment for vaccination status (not, fully ≥6 months prior, fully <6 months prior), boosted, and cancer diagnosis. *b* This is model is the same as the primary model with the addition of a covariable for "known past medical history" which is determined by the presence of at least one ICD-10 code in the UHealth system. *c* The is the unadjusted model for the 1:4 (received REGEN-COV:not) propensity-matched cohort. The propensity score used for matching was created with the following covariables: vaccination status, boosted, age, gender, race/ ethnicity, payor, knowledge of past medical history, cancer diagnosis, and individual Elixhauser comorbidities. Patients were matched 1:4 without replacement using a caliper distance of 25% of the standard deviation of all propensity scores. The matched cohort is described in S1 Table. *d* This is the overfit model including the following covariables: vaccination status, boosted, age, gender, race/ethnicity, payor, knowledge of past medical history, and individual Elixhauser comorbidities. *e* This is the same overfit model as in (d) with the cohort restricted to include only patients with "known past medical history". *f* This is the primary model, but restricted to patients who visited (hospital or emergency department) UHealth within the year prior to testing.

suggested an association of poorer outcomes with REGEN-COV use during Omicron versus Delta, or reduced REGEN-COV effectiveness. It is possible, therefore, that with a larger sample size, this increased odds would have been statistically significant.

To our knowledge, ours is the first assessment of the clinical effectiveness of REGEN-COV in the context of the Omicron vs Delta SARS-CoV-2 variants. To accomplish this, we capitalized on a uniqueness of the pandemic in Miami, Florida—that Delta infections were essentially absent when the initial Omicron surge occurred. This reality allowed us to reliably compare outcomes of Omicron to Delta infections without full population genotyping. However, our study has limitations. First, our data come from a single health system which may result in poor external generalizability and misclassification of our exposure (REGEN-COV) or our outcomes (hospitalization, ED presentations) if patients accessed care outside UHealth. However, we have no reason to expect this to occur differentially during the Delta and Omicron periods. Moreover, the robustness of our results to sensitivity analyses limited to patients known to access care within our system (and, therefore, more likely to receive REGEN-COV and/or present to the hospital/ED at UHealth) suggests the impact of this bias was small. Exposure misclassification also may have occurred for a small minority of patients who were infected with Delta during the Omicron period. And, while we expect Omicron subvariant BA.4 or BA.5 circulation was low during our study period, we did not genotype patients and low levels may have been present; these newer subvariants appear better able to escape

neutralizing antibodies [13] and, therefore, may differentially respond to REGEN-COV. Our study was also limited by our definitions. We defined COVID-19 infection as SARS-CoV-2 test positivity; a subset was likely incidentally found to be positive and did not exhibit COVID-19 symptoms, lessening both their likelihood of REGEN-COV receipt and their need for hospitalization/ED presentation. Similarly, we included all hospitalizations/ED presentations without knowing if they were due to COVID-19-associated illness. Third, as aforementioned, our study was limited by sample size which may have reduced our ability to detect a differential association of REGEN-COV with Omicron vs Delta. Fourth, we did not have access to data on deaths external to our healthcare system and could not, therefore, evaluate death without ED presentation/hospitalization at UHealth as an outcome.

Finally, our study is likely affected by residual confounding. We did not have data on time-dependent environmental factors (e.g., local heard immunity or vaccination rates) or patient-specific factors (e.g., recent COVID-19 illness with possible immunity) which may have affected our results. The most important source of residual confounding is likely confounding by indication. REGEN-COV's EUA and the guidance for its use focus on administration for individuals at high-risk of progression to severe disease. As such, those patients who received REGEN-COV were, by definition, more likely to progress to hospitalization. While adjustment for patient-level differences in risk can mitigate against this confounding, our sample size and the availability of complete information about the chronic illnesses of all cohort patients may have limited our ability to do so adequately. We see the impact of this residual confounding most notably in the increased (albeit not consistently statistically significantly) odds of hospitalization with the administration of REGEN-COV (vs not). As there is no reason to expect that REGEN-COV *causes* worse outcomes, this is assuredly a result of residual confounding. This explanation is supported by the fact that the odds of hospitalization associated with REGEN-COV use decreases in sensitivity models which more completely adjust for possible confounding; specifically, in the models inclusive of all available patient factors, REGEN-COV use was associated with a lower odds of hospitalization.

We expect this residual confounding is less likely to impact our primary findings of the differential association of REGEN-COV use with outcomes in Omicron vs Delta than it is the association of REGEN-COV use itself with outcomes. The interaction term, from which the differential association is assessed, would be subject to residual confounding only if patients receiving REGEN-COV (vs not) differed in the Delta vs Omicron periods. While patients who received REGEN-COV in the Omicron period were slightly older (median [IQR]: 62 [48–71] vs 56 [45–66], p = 0.006), both groups skewed to <65 years-old (a trigger for REGEN-COV use) and they had similar likelihood of comorbid disease (66% vs 68%, p = 0.84; S2 Table). However, we cannot ensure that unmeasured differences do not persist as REGEN-COV use was discouraged during Omicron, thereby potentially limiting use to a unique subset of SARS-CoV-2 positive patients.

Within the limitations of this study's power to detect a difference, we found no clear reduction in the apparent clinical effectiveness of REGEN-COV treatment for SARS-CoV-2 infection to prevent severe disease with the Omicron vs the Delta variant. While the EUA for REGEN-COV has been rescinded, our study reminds us of the need for in-human evaluations for new therapies as outcomes may differ from pre-clinical in vitro and in vivo studies.

## Supporting information

**S1 Table. Propensity matched cohort characteristics by REGEN-COV receipt.**
(DOCX)

**S2 Table. Characteristics of patients who received REGEN-COV by time period.**
(DOCX)

## Author Contributions

**Conceptualization:** Hayley B. Gershengorn, Samira Patel, Tanira Ferreira, Dipen J. Parekh, Bhavarth Shukla.

**Data curation:** Samira Patel.

**Formal analysis:** Hayley B. Gershengorn, Samira Patel, Sankalp Das.

**Investigation:** Hayley B. Gershengorn.

**Methodology:** Hayley B. Gershengorn, Samira Patel.

**Project administration:** Hayley B. Gershengorn.

**Supervision:** Hayley B. Gershengorn.

**Validation:** Hayley B. Gershengorn.

**Visualization:** Hayley B. Gershengorn.

**Writing – original draft:** Hayley B. Gershengorn.

**Writing – review & editing:** Samira Patel, Tanira Ferreira, Sankalp Das, Dipen J. Parekh, Bhavarth Shukla.

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
