## [Decision Letter · Decision Letter 0]

26 Aug 2022

PONE-D-22-20961The Clinical Effectiveness of REGEN-COV in SARS-CoV-2 Infection with Omicron Versus Delta VariantsPLOS ONE

Dear Dr. Hayley Beth Gershengorn,

Thank you for submitting your manuscript to PLOS ONE. After careful consideration, we feel that it has merit but does not fully meet PLOS ONE’s publication criteria as it currently stands. Therefore, we invite you to submit a revised version of the manuscript that addresses the points raised during the review process.

 We appreciate your efforts for the interesting study. However, there are some points raised by the reviewers and need to be clarified. Please carefully respond to the reviewers' comments and suggestions.

We look forward to receiving your revised manuscript.

Kind regards,

Vipa Thanachartwet, M.D.

Academic Editor

PLOS ONE

Journal Requirements:

"HBG was a member of a scientific advisory board for COVID therapeutics for Gilead Sciences, Inc.

The other authors have declared that no competing interests exist. "

Reviewers' comments:

Reviewer's Responses to Questions

**Comments to the Author**

1. Is the manuscript technically sound, and do the data support the conclusions?

Reviewer #1: Yes

Reviewer #2: Yes

2. Has the statistical analysis been performed appropriately and rigorously? 

Reviewer #1: Yes

Reviewer #2: Yes

3. Have the authors made all data underlying the findings in their manuscript fully available?

Reviewer #1: Yes

Reviewer #2: Yes

4. Is the manuscript presented in an intelligible fashion and written in standard English?

Reviewer #1: Yes

Reviewer #2: Yes

5. Review Comments to the Author

Reviewer #1: This study provides interesting and huge amount of data about CASIRIVIMAB/IMDEVIMAB (REGENCOV) use in a different cohort of patients: Delta and Omicron affected patients. It is well written and the statistical analysis is clear and linear.

I have minor revisions and questions to be cleared:

- which dosage of REGENCOV has been used treating Delta and then treating Omicron?

- Omicron lineage has been categorized under subtypes? BA1?BA2 etc etc

- Less hospitalizations were observed during Omicron and for those receiving REGENCOV. Don't you think It could be explained by the different virulence of Omicron rather than Delta?

- FDA authorized the use of REGENCOV for your Omicron affected patients?

- Ambulatory (outpatient) REMDESIVIR was authorized in those months (between 2021 and 2022)? Do you regisered its use in case?

- Why in Delta period about 30% (32%) of patients did not receive REGENCOV? Are all patients without criteria?

- Beside hospitalization rate, do you have mortality data?

- I would cite this real-life work on mABS Falcone M, Tiseo G, Valoriani B, Barbieri C, Occhineri S, Mazzetti P, Vatteroni ML, Suardi LR, Riccardi N, Pistello M, Tacconi D, Menichetti F. Efficacy of Bamlanivimab/Etesevimab and Casirivimab/Imdevimab in Preventing Progression to Severe COVID-19 and Role of Variants of Concern. Infect Dis Ther. 2021 Dec;10(4):2479-2488.

- During these months under study, Sotrovimab was in used?

Reviewer #2: Dr. Gershengorn and colleagues provide a retrospective review of their real-world clinical experience with the use of the monoclonal Ab REGEN-COV during the Delta and Omicron surges at their single center institution. This analysis seeks to answer an important question given concerns for lack of in vitro activity of the Mab for the Omicron lineage variants. This description of their experience attempts to address the relative effectiveness of this treatment during these two periods within the limits of the retrospective and observational nature of the data. The authors do a fairly robust job with their sensitivity analyses and in their discussion of highlighting the limitations of their results, although I agree with their contention that a lack of power to show a difference may be the most likely explanation of their results. A few questions or comments that could be addressed which could potentially enhance this contribution would include the following:

Major Comments:

1. Is there any data on what the rate of other outpatient Rx (other Mabs, short course IV RDV, oral antivirals such as oral nirmatrelvir/ritonavir or molunipiravir) was during these two time periods, especially in the Omicron wave? These were becoming available particularly at the end of this period and would be important to describe the rate of their use both in the group that received REGEN-COV as well as those who did not as this could impact outcomes.

2. Was there any concomitant use of the monoclonal Ab Sotrovimab during the Omicron surge? This might be an interesting sensitivity analysis to look at whether a comparable group treated with a Mab which in vitro has activity against Omicron showed any difference in outcomes.

3. Was there any shortage or limitations of the availability or use of Mab during either surge? If so, how was the scarce resource allocated and what impact would that have had on patient selection? As the authors mention, there is likely residual confounding present and it is possible that particularly with the Omicron surge that providers' threshold or selection bias for whom they treated with REGEN-COV despite emerging evidence it might lack in vitro activity could explain or impact these results.

4. The original trials that showed the impact of Mab on reducing hospitalization or ED visits was in an unvaccinated population so we have much less robust data on the relative and absolute magnitude of benefit in a fully vaccinated, partially vaccinated, and/or boosted population. In addition, there is data to suggest (including that presented by the authors) that Omicron led to lower severity disease and less hospitalization relative to Delta, whether due to intrinsic viral factors or other causes. These may be important factors to highlight in the discussion of the limitations and potential lack of power to detect a true difference if it exists.

5. It might be useful for the authors to do a sensitivity analysis that statistically attempt to calculate what level of association or impact an unmeasured confounder might need to to explain their results. There are different statistical techniques that can be used for this. In studies with a positive result, calculating an E-value can be one way to get at this (see this reference from JAMA https://jamanetwork.com/journals/jama/fullarticle/2723079). In this case of a negative result showing no association, this or other tools might be able to estimate this impact if feasible. Alternatively, some estimate of what type of power (i.e. how many hospitalizations or ED visits would be needed to show a difference if such exists) they ended up could be useful to frame these results for the reader.

6. For figure 1a-d, would suggest that all of these could be combined into one graph, possibly with vertical bar graphs that list the 4 outcomes (prob of hospitalization at 15d and 30d and ED presentation within 15d and 30d) grouped by No REGEN-COV and REGEN-COV receipt. This would save space and provide one table that could compare outcomes for the two groups.

Minor Comments:

1. In line 42 of the introduction, the Delta period is listed as summer 2022 but this would likely be better described as summer-fall 2021.

6. PLOS authors have the option to publish the peer review history of their article (what does this mean?). If published, this will include your full peer review and any attached files.

Reviewer #1: No

Reviewer #2: **Yes: **James Cutrell

---

## [Author Response · Author response to Decision Letter 0]

20 Sep 2022

* PLEASE SEE THE UPLOADED DOCUMENT ENTITLED "RESPONSES TO REVIWER COMMENTS" FOR A CLEANER VERSION (INCLUDING FIGURES) *

We appreciate the opportunity to respond to the reviewer comments. Please see our point-by-point responses indented below each comment.

Reviewer #1: This study provides interesting and huge amount of data about CASIRIVIMAB/IMDEVIMAB (REGENCOV) use in a different cohort of patients: Delta and Omicron affected patients. It is well written and the statistical analysis is clear and linear.

I have minor revisions and questions to be cleared:

- which dosage of REGENCOV has been used treating Delta and then treating Omicron?

The dosage of REGEN-COV that was used is casirivimab 600mg / imdevimab 600mg. This has now been included in the manuscript Methods (lines 58-59): “Subjects were stratified into those who received REGEN-COV (casirivimab 600mg / imdevimab 600mg intravenously once) within 72h of testing positive and those who did not;[2]”.

- Omicron lineage has been categorized under subtypes? BA1?BA2 etc etc

We appreciate this question, especially in light of potentially different clinical characteristics and response to treatments/vaccination of the different subvariants of Omicron. However, we did not genotype our cases and, thus, cannot directly answer the reviewer’s question. Based on the overall trends around the country, we expect that our Omicron cases were composed, largely, of BA1, BA2, and possibly BA3; we expect little BA4 or BA5.

We have now added this to the limitations section (lines 206-209): “And, while we expect Omicron subvariant BA.4 or BA.5 circulation was low during our study period, we did not genotype patients and low levels may have been present; these newer subvariants appear better able to escape neutralizing antibodies [13] and, therefore, may differentially respond to REGEN-COV.”

- Less hospitalizations were observed during Omicron and for those receiving REGENCOV. Don't you think It could be explained by the different virulence of Omicron rather than Delta?

As the reviewer notes, there is evidence that Omicron is less virulent than Delta (e.g., PMID: 35918098). However, our model structure includes an interaction term evaluating the differential association of REGEN-COV with hospitalization in Omicron vs Delta; as such, it “adjusts” for this baseline difference in odds of hospitalization with Omicron by comparing Omicron patients to other Omicron patients (and vice-versa for Delta). The terminology “difference-in-differences” is often used for this type of evaluation.

- FDA authorized the use of REGENCOV for your Omicron affected patients?

As mentioned in the Introduction (lines 32-40), there was an Emergency Use Authorization for REGEN-COV by the FDA for all high-risk COVID-19 patients granted on November 21, 2020; on January 24, 2022 (after the end date of our study, January 7, 2022) the FDA revised this EUA “to exclude geographic regions where… infection or exposure is likely due to a variant that is non-susceptible to REGEN-COV” (page 4, lines 39-40). As such, the EUA was in effect throughout the duration of our study and REGEN-COV was readily available for prescribers in our system.

As of August 26, 2022, the website www.regencov.com still makes the following statement pertaining to that January 24, 2022 EUA revision: “With this EUA revision, REGEN-COV is not currently authorized for use in any U.S. states, territories or jurisdictions, since Omicron is currently the dominant variant across the United States. REGEN-COV remains an investigational drug and is not approved for any indication.”

- Ambulatory (outpatient) REMDESIVIR was authorized in those months (between 2021 and 2022)? Do you regisered its use in case?

Remdesivir was not available for outpatients at UHealth during this time. To the best of our knowledge, it was similarly not available elsewhere in South Florida at the time. We have now explicitly noted this on lines 62-65 of the Methods section: “Notably, there was no access to Sotrovimab at UHealth during this period; similarly, Remdesivir was not available for administration to outpatients and neither nirmatrelvir/ritonavir nor molunipiravir were prescribed for any cohort patients.”

Please see our response to reviewer 2’s major comment 1 below for more information about all outpatient COVID therapies.

- Why in Delta period about 30% (32%) of patients did not receive REGENCOV? Are all patients without criteria?

Thank you for this question. First, to be clear, the data presented in Table 1 (unless otherwise noted) is the count and column percentage. As such, the 32% refers to the fact that 32% of all patients who did not receive REGEN-COV were from the Delta period (68% were from the Omicron period). To make this more clear we have added a footnote to Table 1: “a percentages depict the fraction of each population (column percentages) unless otherwise indicated.”

The percentage of patients who did not receive REGEN-COV in both the Delta and Omicron periods is actually much higher than 32%: Delta: 1870/(213+1870) = 90%; Omicron: 4045/(156+4045) = 96%. These high percentages have face validity for us as many patients who test positive exhibit no/mild symptoms (so they/their providers may not consider REGEN-COV) and/or are not in the high-risk group. While there is little in the literature on the usage rates of REGEN-COV, one single study from Israel (PMID: 35918340) included a full cohort of 162,795 patients testing positive for COVID-19 from July 1 – Dec 8, 2021, of whom 306 received REGEN-COV between Sept 19 – Dec 8, 2021 (when it was available). Assuming a constant rate of test positivity (as actual data on positive cases before and after REGEN-COV was available are not provided), this study would include 7122 patients per week and the number receiving REGEN-COV (when available) would be 27 per week; as such, an estimated >99% ([7122-27]/7122) would not have received REGEN-COV when it was available. 

- Beside hospitalization rate, do you have mortality data?

Unfortunately, we only have access to the death data for patients who die as in-patients (while hospitalized) at our facility; we do not have a linkage to death data outside our health system. However, we agree that this would have been a nice outcome to have access to as it would have been less subject to outcome misclassification than our single health system-based outcomes are. We now note this in our limitations section (lines 215-217): “Fourth, we did not have access to data on deaths external to our healthcare system and could not, therefore, evaluate death without ED presentation/hospitalization at UHealth as an outcome.”

- I would cite this real-life work on mABS Falcone M, Tiseo G, Valoriani B, Barbieri C, Occhineri S, Mazzetti P, Vatteroni ML, Suardi LR, Riccardi N, Pistello M, Tacconi D, Menichetti F. Efficacy of Bamlanivimab/Etesevimab and Casirivimab/Imdevimab in Preventing Progression to Severe COVID-19 and Role of Variants of Concern. Infect Dis Ther. 2021 Dec;10(4):2479-2488.

Thank you for alerting us to this publication which we have now referenced in a new sentence in the Discussion section (lines 182-184), “And, prior clinical evaluations assessing REGEN-COV’s effectiveness across variants predated both Delta and Omicron emergence [11].” 

- During these months under study, Sotrovimab was in used?

Despite an existing FDA EUA for Sotrovimab, we did not have access to it at UHealth until January 7, 2022 (last day of our study period). We have now explicitly noted that Sotrovimab was not available in the Methods section (lines 62-65): “Notably, there was no access to Sotrovimab at UHealth during this period; similarly, Remdesivir was not available for administration to outpatients and neither nirmatrelvir/ritonavir nor molunipiravir were prescribed for any cohort patients.”

Reviewer #2: Dr. Gershengorn and colleagues provide a retrospective review of their real-world clinical experience with the use of the monoclonal Ab REGEN-COV during the Delta and Omicron surges at their single center institution. This analysis seeks to answer an important question given concerns for lack of in vitro activity of the Mab for the Omicron lineage variants. This description of their experience attempts to address the relative effectiveness of this treatment during these two periods within the limits of the retrospective and observational nature of the data. The authors do a fairly robust job with their sensitivity analyses and in their discussion of highlighting the limitations of their results, although I agree with their contention that a lack of power to show a difference may be the most likely explanation of their results. A few questions or comments that could be addressed which could potentially enhance this contribution would include the following:

Major Comments:

1. Is there any data on what the rate of other outpatient Rx (other Mabs, short course IV RDV, oral antivirals such as oral nirmatrelvir/ritonavir or molunipiravir) was during these two time periods, especially in the Omicron wave? These were becoming available particularly at the end of this period and would be important to describe the rate of their use both in the group that received REGEN-COV as well as those who did not as this could impact outcomes.

We appreciate this point (which is similar to those made by Reviewer 1 above pertaining to the use of outpatient Remdesivir and, separately, Sotrovimab). Despite use becoming more available at this time, in our system and, to our knowledge, in South Florida at large, availability during our study period for use in the outpatient setting was low/non-existent. 

Remdesivir was not available for outpatients at UHealth during this time (22 of our cohort patients had Remdesivir ordered with 72h of testing positive, but all after being hospitalized). To the best of our knowledge, it was similarly not available elsewhere in South Florida at the time. 

Despite an existing FDA EUA for Sotrovimab, we did not have access to it at UHealth until January 7, 2022 (last day of our study period). 

Similarly, neither nirmatrelvir/ritonavir nor molunipiravir was ordered for any cohort patient.

To make readers aware of this important information, we have now explicitly noted this on lines 62-65 of the Methods section: “Notably, there was no access to Sotrovimab at UHealth during this period; similarly, Remdesivir was not available for administration to outpatients and neither nirmatrelvir/ritonavir nor molunipiravir were prescribed for any cohort patients.”

2. Was there any concomitant use of the monoclonal Ab Sotrovimab during the Omicron surge? This might be an interesting sensitivity analysis to look at whether a comparable group treated with a Mab which in vitro has activity against Omicron showed any difference in outcomes.

We appreciate this comment which was also mentioned by reviewer 1. As mentioned above, despite an existing FDA EUA for Sotrovimab, we did not have access to it at UHealth until January 7, 2022 (last day of our study period). We have now explicitly noted that Sotrovimab was not available in the Methods section (lines 62-65): “Notably, there was no access to Sotrovimab at UHealth during this period; similarly, Remdesivir was not available for administration to outpatients and neither nirmatrelvir/ritonavir nor molunipiravir were prescribed for any cohort patients.”

While it would have been interesting, as suggested, to evaluate the relative associations of REGEN-COV vs Sotrovimab with outcomes during Omicron, due to its non-use, we cannot perform this analysis. That said, its non-use is of benefit here as well as there is no confounding by Sotrovimab exposure.

3. Was there any shortage or limitations of the availability or use of Mab during either surge? If so, how was the scarce resource allocated and what impact would that have had on patient selection? As the authors mention, there is likely residual confounding present and it is possible that particularly with the Omicron surge that providers' threshold or selection bias for whom they treated with REGEN-COV despite emerging evidence it might lack in vitro activity could explain or impact these results.

Thank you for raising this important issue of medication shortages which have clearly plagued us during this pandemic. Luckily, although we did experience shortages of Sotrovimab after initially procuring it, we did not experience shortages of REGEN-COV.

4. The original trials that showed the impact of Mab on reducing hospitalization or ED visits was in an unvaccinated population so we have much less robust data on the relative and absolute magnitude of benefit in a fully vaccinated, partially vaccinated, and/or boosted population. In addition, there is data to suggest (including that presented by the authors) that Omicron led to lower severity disease and less hospitalization relative to Delta, whether due to intrinsic viral factors or other causes. These may be important factors to highlight in the discussion of the limitations and potential lack of power to detect a true difference if it exists.

We fully agree that the challenges of vaccination (rates changing over time as well as waning immunity from prior vaccinations) may have affected the rates of ED presentation and hospitalization during Omicron differently than Delta. While imperfect, our inclusion of vaccination (including time from inoculation) and booster status as confounders in the models seeks to address this issue. 

The issue of differential illness severity intrinsic to Omicron vs Delta is, as noted by this reviewer and Reviewer 1 above, becoming more well-known (e.g., PMID: 35918098). As noted above in response to Reviewer 1’s similar comment, our model structure including an interaction term evaluates the differential association of REGEN-COV with hospitalization in Omicron vs Delta; as such, it “adjusts” for this baseline difference in odds of hospitalization with Omicron by comparing Omicron patients to other Omicron patients (and vice-versa for Delta). The terminology “difference-in-differences” is often used for this type of evaluation. 

The manner in which this lower overall rate of severe illness may impact our findings is, as this reviewer notes, in its potential impact on the lack of power of our models. Determining the power of a logistic regression model (or, determining the number of variables one can include in a logistic regression model before sufficient power is lost) is a matter of ongoing debate. However, what is known is that having more events per variable (in this case, hospitalizations per variable) results in a better fit model (Austin and Steyerberg, Stat Methods Med Res, 2017. PMID: 25411322). As such, if the event rate declined as we entered Omicron, we would have lost power in our model. This important point is now made in lines 187-190 of the Discussion: “As power for associative studies depends on the number of events, the fact that rates of severe outcomes overall (due to virulence differences [12], vaccination uptake, or something else) declined during the Omicron wave may partly explain our estimate imprecision.”

5. It might be useful for the authors to do a sensitivity analysis that statistically attempt to calculate what level of association or impact an unmeasured confounder might need to to explain their results. There are different statistical techniques that can be used for this. In studies with a positive result, calculating an E-value can be one way to get at this (see this reference from JAMA https://jamanetwork.com/journals/jama/fullarticle/2723079). In this case of a negative result showing no association, this or other tools might be able to estimate this impact if feasible. Alternatively, some estimate of what type of power (i.e. how many hospitalizations or ED visits would be needed to show a difference if such exists) they ended up could be useful to frame these results for the reader.

We appreciate this interesting and thought-provoking idea. After reading up on the E-value, it is clear that it was designed (as noted by the reviewer) to be used when an association is found to determine the magnitude of residual confounding that would be required to nullify that association. However, on the www.evalue-calculator.com website, it is noted that one can “consider the confounding strength capable of moving the observed association to any other value (e.g. … increasing a near-null observed association to a value that is of scientific importance).”

We must note, however, that it is an interaction term whose odds-ratio’s “movement” away from the null (OR [95% CI]: 2.31 [0.76,6.92] � x.xx [1.00001,x.xx]) we would be seeking. And, we were unable to identify any literature on the validity of (or how one would construct) an E-value for an interaction term. Moreover, we did not come across any other non-E-value based techniques that would accomplish what this reviewer suggests for a null result for an interaction term. As such, we are not able to perform this interesting analysis. However, if the reviewer or editors are aware of a technique that is appropriate for this situation, we would be more than happy to include it.

6. For figure 1a-d, would suggest that all of these could be combined into one graph, possibly with vertical bar graphs that list the 4 outcomes (prob of hospitalization at 15d and 30d and ED presentation within 15d and 30d) grouped by No REGEN-COV and REGEN-COV receipt. This would save space and provide one table that could compare outcomes for the two groups.

We appreciate the concern raised by the reviewer that it may be hard to compare the differences in all 4 outcomes (hospitalization within 30d and 15d; ED presentation within 30d and 15d) in the figure panels as they were included. And, we have created the bar graph as suggested (see Figure A below). We feel that lost in this representation, however, is the difference (Omicron vs Delta) in the differences (REGEN-COV vs no REGEN-COV) that the more typical margin probability plots (our original Figure 1 panels) show as lines. 

The interaction term in our models asks, essentially: are the differences in the slopes depicted in our initial Figure 1 panels statistically significant? As such, the original graph, we believe, is a better depiction of our results. We understand that it may be difficult to appreciate this with the 4 panels uploaded separately and, thus, we have now uploaded them as a single 4-paneled figure for the reviewer and the editors’ to review (replicated below as Figure B).

 

Figure A. Alternative Figure 1 (Bar Graph)

Figure B. Combined Original Figure 1

Minor Comments:

1. In line 42 of the introduction, the Delta period is listed as summer 2022 but this would likely be better described as summer-fall 2021.

Thank you very much. This error has been corrected.

---

## [Decision Letter · Decision Letter 1]

26 Oct 2022

PONE-D-22-20961R1The Clinical Effectiveness of REGEN-COV in SARS-CoV-2 Infection with Omicron Versus Delta VariantsPLOS ONE

Dear Dr. Gershengorn,

Thank you for submitting your manuscript to PLOS ONE. After careful consideration, we feel that it has merit but does not fully meet PLOS ONE’s publication criteria as it currently stands. Therefore, we invite you to submit a revised version of the manuscript that addresses the points raised during the review process.

 The authors have made a careful revision to the manuscript. However, there are some minor points that are required to define clearly. Please carefully respond to the reviewer’ comments and suggestions.

We look forward to receiving your revised manuscript.

Kind regards,

Vipa Thanachartwet, M.D.

Academic Editor

PLOS ONE

Journal Requirements:

Reviewers' comments:

Reviewer's Responses to Questions

**Comments to the Author**

1. If the authors have adequately addressed your comments raised in a previous round of review and you feel that this manuscript is now acceptable for publication, you may indicate that here to bypass the “Comments to the Author” section, enter your conflict of interest statement in the “Confidential to Editor” section, and submit your "Accept" recommendation.

Reviewer #1: All comments have been addressed

Reviewer #2: All comments have been addressed

2. Is the manuscript technically sound, and do the data support the conclusions?

Reviewer #1: Yes

Reviewer #2: Yes

3. Has the statistical analysis been performed appropriately and rigorously? 

Reviewer #1: Yes

Reviewer #2: Yes

4. Have the authors made all data underlying the findings in their manuscript fully available?

Reviewer #1: Yes

Reviewer #2: Yes

5. Is the manuscript presented in an intelligible fashion and written in standard English?

Reviewer #1: Yes

Reviewer #2: Yes

6. Review Comments to the Author

Reviewer #1: I appreciate the answers the authors have given. Despite the missing data and the difficulties to satisfy the request of the reviewers I think this work could be now available to be published. Maybe it could be better to avoid to infere statistical data about efficacy or not given the missing data. I think it would be more appropriate just to describe the experience.

Reviewer #2: Overall, Dr. Gershengorn and colleagues provide a robust retrospective analysis of their clinical experiences with REGN-CoV Mab therapy across the respective Delta and Omicron periods at their institution. Within the limits of a retrospective observational study, they do a rigorous job of controlling for potential confounding to analyze for any differential effectiveness. The authors have helpfully clarified several important aspects in the methods about the use of any other outpatient COVID-19 therapeutics. They have also added additional important comments in the limitations section of their discussion.

A few additional comments that I would add with regards to the paper are as follows:

1. I still think that the most likely explanation for the findings of no differential effectiveness is due to inadequate power to detect a difference due to a lower number of hospitalizations and ED visits, particularly in the Omicron period. The authors adequately discuss and explore this limitation in their discussion; however, I think that at least a passing comment in the abstract is also appropriate since some people will not read the full paper. So a statement such as the following in the abstract conclusion would be useful: "Within the limitations of this study's power to detect a difference, we identified no differential effectiveness..."

2. I would remove the final sentence of the conclusion (lines 245-247) as I don't think that it is warranted based on the available data. While it is generally true that having rigorous RCTs is always preferable to relying on in vitro or clinical data, the reality in the COVID-19 pandemic is that the rapid emergence and evolution over time of clinical variants has rendered it impractical to conduct new adequately powered RCTs for each new emergent COVID-19 variant Therefore, the FDA has had to rely on this type of indirect data to determine the EUA status and recommendations for Mab therapy use. The final sentence seems to imply that based on this data we should have continued to use REGEN-CoV until we have RCT trial data to suggest otherwise; however, there is enough uncertainty and limitations to this data that I don't think that position can be fully supported in the face of consistent and compelling in vitro data showing a loss of activity against the newer Omicron lineage variants. I would just end the conclusion with the next to last sentence saying something similar to the abstract: "Within the limitations of this study's power to detect a difference, we found no clear reduction in the apparent clinical effectiveness..."

3. I agree with the authors that their new 4-panel version of Figure 1 adequately addresses the prior concerns.

7. PLOS authors have the option to publish the peer review history of their article (what does this mean?). If published, this will include your full peer review and any attached files.

Reviewer #1: **Yes: **Lorenzo Roberto Suardi

Reviewer #2: **Yes: **James Cutrell

---

## [Author Response · Author response to Decision Letter 1]

27 Oct 2022

We thank the reviewers and editors again for their thorough review of our work. Please see below for a point-by-point response to the remaining concerns.

Reviewer #1: I appreciate the answers the authors have given. Despite the missing data and the difficulties to satisfy the request of the reviewers I think this work could be now available to be published. Maybe it could be better to avoid to infere statistical data about efficacy or not given the missing data. I think it would be more appropriate just to describe the experience.

Thank you for your comments. We agree that data missingness is a limitation but hope we have appropriately described it such that readers can correctly interpret our findings.

Reviewer #2: Overall, Dr. Gershengorn and colleagues provide a robust retrospective analysis of their clinical experiences with REGN-CoV Mab therapy across the respective Delta and Omicron periods at their institution. Within the limits of a retrospective observational study, they do a rigorous job of controlling for potential confounding to analyze for any differential effectiveness. The authors have helpfully clarified several important aspects in the methods about the use of any other outpatient COVID-19 therapeutics. They have also added additional important comments in the limitations section of their discussion.

Thank you.

A few additional comments that I would add with regards to the paper are as follows:

1. I still think that the most likely explanation for the findings of no differential effectiveness is due to inadequate power to detect a difference due to a lower number of hospitalizations and ED visits, particularly in the Omicron period. The authors adequately discuss and explore this limitation in their discussion; however, I think that at least a passing comment in the abstract is also appropriate since some people will not read the full paper. So a statement such as the following in the abstract conclusion would be useful: "Within the limitations of this study's power to detect a difference, we identified no differential effectiveness..."

We appreciate this concern and have now amended the Abstract’s conclusions to include the proposed phrase. 

2. I would remove the final sentence of the conclusion (lines 245-247) as I don't think that it is warranted based on the available data. While it is generally true that having rigorous RCTs is always preferable to relying on in vitro or clinical data, the reality in the COVID-19 pandemic is that the rapid emergence and evolution over time of clinical variants has rendered it impractical to conduct new adequately powered RCTs for each new emergent COVID-19 variant Therefore, the FDA has had to rely on this type of indirect data to determine the EUA status and recommendations for Mab therapy use. The final sentence seems to imply that based on this data we should have continued to use REGEN-CoV until we have RCT trial data to suggest otherwise; however, there is enough uncertainty and limitations to this data that I don't think that position can be fully supported in the face of consistent and compelling in vitro data showing a loss of activity against the newer Omicron lineage variants. I would just end the conclusion with the next to last sentence saying something similar to the abstract: "Within the limitations of this study's power to detect a difference, we found no clear reduction in the apparent clinical effectiveness..."

We agree that it is worth restating the potential limitation introduced by our sample size in the concluding paragraph and have, thus, revised the first sentence as suggested (same as in the Abstract) to do just this. 

As for the last sentence, it was not our intention to suggest that clinical trials, per se, be done; but, rather, that evaluations in humans (vs in vitro or pre-clinical in vivo work) is important. As such, our choice of wording (“clinical trials”) was in error, and we’ve now changed it to “in-human evaluations”. Similarly, it was not our intent to state that REGEN-COV itself should be studied in an in-human evaluation at this point; rather, our goal was to point out the value of our study in calling attention to the fact that pre-clinical and clinical studies often have divergent findings. To try to make clearer that this comment was not intended to apply to only REGEN-COV per se, we have added the phrase “for new therapies”. With these changes, we have elected to leave the last sentence in the paper as we think this making point—the importance of studying humans for therapies to be used on humans—is valuable. We hope these changes make the sentence more appropriate (palatable?) from the perspective of the reviewer.

3. I agree with the authors that their new 4-panel version of Figure 1 adequately addresses the prior concerns.

Thank you.

---

## [Decision Letter · Decision Letter 2]

23 Nov 2022

The Clinical Effectiveness of REGEN-COV in SARS-CoV-2 Infection with Omicron Versus Delta Variants

PONE-D-22-20961R2

Dear Dr. Gershengorn,

We’re pleased to inform you that your manuscript has been judged scientifically suitable for publication and will be formally accepted for publication once it meets all outstanding technical requirements.

Kind regards,

Vipa Thanachartwet, M.D.

Academic Editor

PLOS ONE

Additional Editor Comments (optional):

The authors have revised all comments raised by the reviewers.

Reviewers' comments:

Reviewer's Responses to Questions

**Comments to the Author**

1. If the authors have adequately addressed your comments raised in a previous round of review and you feel that this manuscript is now acceptable for publication, you may indicate that here to bypass the “Comments to the Author” section, enter your conflict of interest statement in the “Confidential to Editor” section, and submit your "Accept" recommendation.

Reviewer #1: All comments have been addressed

Reviewer #2: All comments have been addressed

2. Is the manuscript technically sound, and do the data support the conclusions?

Reviewer #1: Yes

Reviewer #2: Yes

3. Has the statistical analysis been performed appropriately and rigorously? 

Reviewer #1: Yes

Reviewer #2: Yes

4. Have the authors made all data underlying the findings in their manuscript fully available?

Reviewer #1: Yes

Reviewer #2: Yes

5. Is the manuscript presented in an intelligible fashion and written in standard English?

Reviewer #1: Yes

Reviewer #2: Yes

6. Review Comments to the Author

Reviewer #1: After this second revision, I think the work Is suitable to be published.

I appreciate the effort that have been made in order to clarify the reviewers' doubts

Reviewer #2: Authors have satisfactorily addressed the limitations in their study and the new comments added to the abstract and final paragraph of the discussion more clearly state these.

7. PLOS authors have the option to publish the peer review history of their article (what does this mean?). If published, this will include your full peer review and any attached files.

Reviewer #1: **Yes: **Lorenzo Roberto Suardi

Reviewer #2: No

---

## [Editor Report · Acceptance letter]

25 Nov 2022

PONE-D-22-20961R2 

The Clinical Effectiveness of REGEN-COV in SARS-CoV-2 Infection with Omicron Versus Delta Variants 

Dear Dr. Gershengorn:

I'm pleased to inform you that your manuscript has been deemed suitable for publication in PLOS ONE. Congratulations! Your manuscript is now with our production department. 

Kind regards, 

on behalf of

Associate Professor Vipa Thanachartwet 

Academic Editor

PLOS ONE